# Trends in the Use of Home LTC Services in Large, Medium and Small Municipalities in Italy: Lessons for the Post-COVID-19 Reappraisal

**DOI:** 10.3390/ijerph191912796

**Published:** 2022-10-06

**Authors:** Carlo Lallo, Marta Pasqualini, Cecilia Tomassini

**Affiliations:** 1Department of Economics, University of Molise, Via F. De Sanctis, 86100 Campobasso, Italy; 2Department MEMOTEF, Sapienza University of Rome, Via del Castro Laurenziano 9, 00161 Rome, Italy

**Keywords:** Italy, LTC, care provision, ageing in place, municipality size, regional differences, COVID-19 pandemic

## Abstract

Italian Long-Term Care is considered largely inadequate, and the recent COVID-19 pandemic has dramatically exposed its limitations. Public Home Care Services in particular were revealed as under-financed and unable to cover the potential demand for care from the older population. But does the type of municipality and its geographic location play a role in creating or mitigating unmet demand? This is the first study addressing this research question in Italy. Our hypothesis is that older people’s care preferences and care possibilities may vary between small, medium and metropolitan areas, as will the organisation, funding and availability of services, and the combination will influence (unmet) demand for public home care services. In this paper, using nationally representative survey data collected by the Italian National Statistical Institute in 2003 and 2016, we investigate changes and differences in the use of public and private home care services among people aged 75 or older in Italy by size of the municipality. Our results reveal inequalities in service use between Northern and Southern areas of the country and in particular between metropolitan areas, medium and small municipalities. Such differences reinforce post-pandemic calls for new investment and changes in the design of the Italian Long-Term Care system.

## 1. Introduction

The population of Italy is one of the oldest in the world; according to the Italian National Institute of Statistics (ISTAT), life expectancy in 2021 was 82.32 at birth and 20.29 at 65. The high life expectancy combined with a lowest-low fertility rate (1.24) has accelerated and magnified an ageing process seen in all Western countries. In Italy, the old-age dependency ratio has reached 187.9% in 2022; 23.8% of the population is aged 65 or more, and only 12% is aged 14 or less. While increases in life expectancy represent a positive trend, the changes in the age structure of the population demand a thoughtful evaluation of the current capability of the social and welfare system, leading to the design of new and more effective policies. One area of the system directly affected by the ageing process is Long-Term Care (LTC), comprising a variety of services designed to help people live as independently and safely as possible when they age or can no longer perform activities of daily living (ADL) on their own.

According to ISTAT [1], in 2019, 3.8 million people of all ages had severe limitations in ADL, such as eating, dressing, getting into or out of a bed or chair, taking a bath or shower and using the toilet; they represented 10% of people aged 65+ and 18.3% of those aged 75+. Moreover, around 32% of people aged 65+ and 52% of people aged 75+ were living with severe chronic illness. Again in 2019, around 44% of requests for care, as reported by older people with severe limitations in ADL, remained unfulfilled (ISTAT, 2021). This inadequacy of the Italian public LTC system was ruthlessly highlighted by the COVID-19 pandemic [2,3,4,5,6]. The public expenditure on Italian LTC is polarised between two main targets: institutions, such as nursing homes (in Italian *Case di Riposo*—RSA—and *Residenze Protette*—RP), and cash allowances to support private formal or informal home care (Care Allowance—CA), the latter representing around 55% of all public LTC expenditure [7,8]. The tragic experience of those living in nursing homes, who suffered higher mortality, isolation and distress [4,9], is one of the most important lessons learned in Italy from the pandemic. The frailest, who needed more protection, were eventually the most exposed to the infection. Having concentrated (lacking) LTC services in hospitals and nursing homes had disastrous consequences [2,5] since COVID-19 spread more quickly and more lethally in such institutions, especially in Northern regions [10]. 

Even if institutionalization may be the only choice for a minority of older people with severe health problems, the COVID-19 pandemic showed that structural changes, more investment and a better integration with home care services are required [11,12]. The spread of COVID-19 and related excess of mortality has been favoured, on the one side, by a lack of trained personnel (medical and non-medical) in nursing homes, difficulties in isolating COVID-19-infected residents and by the concurrent isolation of nursing homes from the rest of the health system. On the other side, older people living at home had difficulties in accessing hospitals or in receiving social and health support at home. 

The literature has already introduced the concept of “ageing in place”, defined as the ability to live in one’s own home and community safely, independently and comfortably, regardless of age, income or ability level (e.g., [13,14,15]). Many studies had underlined how promoting public services that can facilitate “a good ageing in place” (public home care services, such as home health care and domestic help) as an alternative to, or in combination with, institutionalization, is the key to ensure a healthy ageing (e.g., [16,17]). However, even before the pandemic, Italian public policies promoting ageing in place were considered largely inadequate [7,18,19,20]. Relative public expenditure on LTC has declined over time since 2005, considering the overall per capita expenditure in real terms [8] and shows remarkable geographic disparities [11,21]; in addition, the Great Recession starting in 2009–2010 had a negative impact on all public expenditure. This has led families in need of LTC to increase informal home care or find a substitute in the private market, often hiring migrant care workers with irregular contracts and a low level of professionalization [22]; at around 700,000, the number of migrant care workers employed in Italy is among the highest. This may be attributed to five factors: general preference for having relatives assisted in their home environment; inadequate supply of public home care services; a relatively easy-to-access cash allowance program for individuals with severe disability (Care Allowance); low salaries for migrant care workers; and finally, progressively shrunken and geographically dispersed families [23]. However, the COVID-19 pandemic showed the severe shortcomings of this approach as well, with care for recipients compromised due to such factors as lack of worker skills, worker vulnerability to infection or recipient inaccessibility due to government policies aimed at restricting the spread of infection [24].

A review of the aims, methods and funding of LTC is urgently required, with a view to redesigning the system; this goal demands the collation of evidence about the current system, its effectiveness or otherwise, and trends in demand. The situation at the national level, briefly summarised above, has been investigated by both policymakers and the scientific literature; however, less is known about potential geographic variation in the use of home care services and the causes of such variation in terms either of municipality size or of geographic location [25,26]. Several factors suggest that this merits investigation. Italy has large regional differences in public resources availability, which, due to decentralisation of health and social public expenditure, has a direct effect on local levels of home care service provision. Meanwhile, increasing demographic differences are evident between metropolitan areas and small municipalities and between southern and northern areas in terms of depopulation and population ageing [27]. The richest northern metropolitan areas attract more and more people from small municipalities and southern areas, mostly young, accelerating the ageing both of small and of southern communities, and dispersing families [28]. Last but not least, populations in metropolitan areas and small municipalities show different preferences with respect to ageing at home or institutionalisation and different relations with their social network [25,26,29,30].

Using ISTAT national samples, in this study, associations are analysed between the use of public or private home care services and the geographic level, namely, the size of the municipality and the region where it is located. Following ISTAT categories, municipalities have been classified in three groups by size of population: (1) less than 10 thousand inhabitants; (2) more than 10 thousand inhabitants; and (3) metropolitan areas. Metropolitan areas include municipalities with more than 250 thousand inhabitants and surrounding small municipalities (Metropolitan centres include: Torino, Milano, Venezia, Genova, Bologna, Firenze, Roma, Napoli, Bari, Palermo, Catania and Cagliari. More info at: https://www.istat.it/it/archivio/129956. Last Access: 28 July 2022).

This research addresses two main questions: (1) how use of home care services in Italy varied between 2003 and 2016, across the so called “Great Recession”, by municipality size and geographic location and (2) how selected explanatory variables (including sex, age, marital status, spouse and kin availability) may interact with use of care services. Understanding the potential correlations between municipality size/geographic location and use of home care services will contribute to the design of better services [31], able at least to mitigate the effects of new pandemic waves. Despite the relevance of geographic variability in the Italian LTC system, to our knowledge, this is the first study investigating this issue using micro-data from two nationally representative surveys.

We constructed three separate logistic regression models on the probability of using home social and health care services and private care, by municipality size and geographic location, and other covariates that have been found crucial in the use of such services in past research, using data collected by ISTAT in 2003 and 2016.

In the next section, the main features of the Italian LTC system are described, explaining the differences between public home health care services and public home social care services, underlining both trends over time and geographic differences in resource availability and in the use of services. The Data and Methods section describes the data and methodology used to answer the research questions. In the last three sections, results (Results) are presented and discussed (Discussion), and finally, some first lessons for post-COVID-19 policies (Conclusions) are suggested.

## 2. Long-Term Care in Italy

Italy has a Mediterranean welfare regime, and LTC is not conceived and organised as a national public comprehensive system but is the result of multiple legislative interventions to integrate different national and regional laws regarding social and health services.

The public LTC system rests on four pillars: Care Allowances, nursing homes, home care health services and home care social services. As pointed out in the introduction, public resources are monopolized by Care Allowances and nursing homes, leaving home care services in a very residual position [32]. The total public LTC expenditure designated to older people was around 1.13% of the Italian GDP in 2016 (it was 1.07% in 2005 [8]).

Care Allowances absorbed around 55% of the total public expenditure on LTC in 2016 (46% in 2005), covering around 1.83 million people (78% of whom were aged 65+), and was designed as a cash allowance programme for individuals with severe disability [8]. The National Institute of Social Security (INPS) manages the Care Allowance, which is financed by general taxation, granting around 522 euros to 13.5% of Italian people aged 65 and over. The benefit is neither means-tested nor subject to accountability on how it is spent and does not vary according to the needs of the recipient.

The second pillar is nursing homes, regionally managed and financed via the National Health System (NHS). The regions are the first-level administrative divisions of the Italian Republic and are autonomous entities within defined powers and budgets. The potential funding of nursing homes is therefore limited by the regional budget and requires co-payment from the users: the lower the economic resources of the Region, the higher the co-payment required. This represents a serious threat to the equality of the system [33]. In 2018, around 295 thousand people, 2.1% of Italians aged 65 and over, were living in nursing homes (a decline from 2.5% in 2009, estimates based on NHS data [34]); however, there were large geographic differences in the availability of the service ranging from 0.7 beds to every 100 people aged 65+ in the southern region of Campania, to 4.4 beds per 100 people aged 65+ in the northern region of South—Tyrol.

With regard to home care services, the focus of this study, the Italian public system offers two distinct types of service. The first is the home health care service (Assistenza Domiciliare Integrata—ADI), managed and funded by the Regions via the NHS. Home health care includes medical and rehabilitation services to assist people with severe disability. According to the latest data from the NHS, in 2018, around 2.7% of people aged 65+ used this home care service. This proportion has increased over time (from 1.29% in 2012), but the time committed by doctors and nursing personnel has decreased from 20 h per year per capita aged 65 and over in 2007 to 17 h in 2013 [35,36,37]. Considering that 10% of people aged 65+ and 18% of people aged 75+ suffer severe limitations in ADL, the coverage level of this service has been described as inadequate [18,38]. Unlike for nursing homes, no co-payments are required, but provision is limited by the regional budget.

Finally, the home social care service (Servizio di Assistenza Domiciliare—SAD) is the last pillar of the Italian public LTC system. Municipalities manage and finance this service, intended to assist older and dependent people. It was severely cut during the Great Recession 2009–2010: total expenditure increased continuously from 5.2 million in 2003 to 7.1 million euros in 2009, but then declined and reached a minimum in 2012 at 6.9 million euros. After 2012, a new increase began, reaching 7.5 million euros in 2018 ([21], estimates based on ISTAT data).

Similarly to ADI, SAD is strongly limited by the municipal budget; municipal expenditure on SAD ranged in 2018 from 6 to 567 euros per capita. There was considerable geographic variation in resources assigned, being greater in the North (around 152 euros per capita) and metropolitan areas (around 110 euros) and lower in the South (around 89 euros) and in small municipalities (around 93 euros) ([21], estimates on ISTAT data).

Evidence suggests that SAD is considerably under-financed and unable to meet the care needs of an increasing aged population with associated limitations in ADL. In 2017, only 1% of people aged 65 and over were assisted by such services, lower than in 2006 (1.8%), whereas the total expenditure per capita (on people aged 65+) increased from 1646 to 2037 euros [35]. Municipalities reacted to the steady increase in numbers of older people in need of care and in intensity of care needed by restricting the economic and health admission criteria and at the same time increasing the co-payment quota required [18,35]. This has led to two apparent paradoxes: first, an increasing share of the population in need of home social care and a decreasing share of the same people accessing the service; second, a decreasing share of older people accessing the service but an increasing per capita expenditure, due to the selection of the frailest. As described in the previous section, three pillars (nursing homes, home health care and social home care) are then unable to cover the requests for care coming from the Italian older population. Only the CA is able to cover about 13% of people aged 65+, a similar percentage to that of people aged 65+ with severe limitations in ADL (10%), even if the size amount of the cash allowance is not very generous (around 522 euros per annum).

It follows that Italian older people prefer to employ migrant care workers (often on an irregular basis) since the CA is the only LTC available measure to cover the costs of a private carer, even it is not sufficient for a full-time professional private care worker with a regular contract (cf. [39]); the latter would require 1267 euros, according to the national collective contract for care workers, signed in 2022 by Trade Unions and the Italian Ministry of Labour (Available online: https://www.lavoro.gov.it/notizie/Pagine/Lavoro-domestico-accordo-sui-minimi-retributivi-2022.aspx. Accessed on 5 September 2022). It is not surprising that only one-third of private care workers in Italy are employed with a regular contract or that most are migrants from Eastern Europe, with low salary expectations [40]. At least 700 thousand migrants are employed in Italy as care workers, whereas, for example, in Germany, Spain and Greece, the equivalent numbers are estimated at around 200 thousand, 223 thousand and 250 thousand, respectively [23,40]. The irregular nature of migrant care workers poses two critical issues, highlighted by the COVID-19 pandemic: first, professional skills are very low, compromising the quality of care they can offer to older people, particularly in emergency situations; second, the irregular nature of their employment (and generally of their status in Italy) limits their own access to public health systems [41]. In addition, during the pandemic, social restrictions and other policies addressed to contain the diffusion of the pandemic, limiting and strictly monitoring the movement of people within and between cities (and, even more, between regions and countries) compromised the ability of migrant care workers to take care of older people [23,42].

To summarise, the Italian home LTC system appears to be primarily based on private (often irregular) care workers, with the public services playing a residual role: this setting was severely affected during the COVID-19 pandemic, demanding new and greater investments in public home care services.

Finally, since the current system is extremely dependent on regional and municipal budgets [25,26,29,30], unveiling the potential correlations between the use of home care services and the size and location of Italian municipalities could provide useful lessons for the process of LTC system redesign after the COVID-19 pandemic, as is indeed the aim of this study.

## 3. Data and Methods

### 3.1. Study Population

Data are drawn from the Family and Social Subjects (FSS) surveys carried out by the Italian Institute of Statistics (ISTAT) in 2003 [43] and 2016 [44]. The FSS surveys are independent nationally representative cross-sectional surveys of a sample of Italian households, collecting information on a range of demographic and socio-economic characteristics. The response rate for each survey ranges between 75 and 80%. The analytic sample we use in this study is restricted to respondents aged 75 and over (since they are more likely to use care services), consisting of 4014 individuals in 2003 and 3314 in 2016.

### 3.2. Outcome Variables

The surveys include several questions on care provision and care receipt. In particular, three measures of care have been considered: the use of health, social and private services. Respondents were asked whether they or any of their family members had received any health or social care in the last 12 months, with possible answers categorised as Yes = 1, No = 0. Additionally, they were asked whether the household usually employs one or more private helpers, and this variable was similarly categorised.

### 3.3. Independent Variables

The main independent variables were the size of the respondent’s municipality, categorised as follows: small (<10,000 inhabitants), medium (10,000–250,000 inhabitants) and large areas (Metropolitan, >250,000 inhabitants), and the macro-region of residence (Northeast; Northwest; Centre; South of Italy; and Islands.

Among control variables, in line with the extensive literature investigating individual characteristics associated with care receipt, we consider gender (Men vs. Women) and marital status (Married/cohabiting; Widowed; Separated/Divorced/Never married). Women have been found to have a higher use of health services [45], but gender differences seem to disappear once health status and socio-economic characteristics are included in the analysis [46]. Marital status and the availability of kin may also be associated with older people’s use of social and health services since older people having a spouse or living in close proximity to children may be more likely to use informal care rather than social services [46]. We include a variable on whether the respondent is childless; he/she has no children living in close proximity (>1 km); or he/she has at least one child living close by (<1 km).

As indicators of socio-economic circumstances, we include educational attainment (secondary or tertiary education vs. compulsory education level) and level of satisfaction with the household’s economic resources at the time of the interview (Enough vs. Not enough). Literature shows how low SES has been found in some contexts (e.g., in The Netherlands [46]) to be associated with higher odds of using formal and informal help. In addition, as an indicator of health, we control for the presence of chronic diseases as this is the only measure of health comparable across the two surveys. Having a chronic disease is likely to be associated with the use of health services and (probably to a lesser extent) with social services.

### 3.4. Statistical Model

To investigate the associations between the demographic and geographic dimensions of the respondent’s area of residence and the use of health, social and private care services, we ran three separate logistic regression models for each type of service used. Models have been stratified by wave (i.e., 2003 and 2016) to detect any differences over time. To ease the interpretation of findings we computed Average Marginal Effects (AMEs), which express how the probability of observing the outcome changes for a unitary increase in a specific independent variable. All the analyses were performed using Stata 16.

## 4. Results

### 4.1. Descriptive Statistics

Table 1 lists weighted descriptive statistics of the two samples.

The proportion of respondents aged 75 and over who had used social services in the previous 12 months was about 2.9% in 2003 and 2.1% in 2016. In addition, health services had been used by 7.6% of the respondents in 2003 and by 9.7% of them in 2016. Finally, about 5.7% (in 2003) and 7.8% (in 2016) reported having a private carer in the last 12 months. About 60% of the sample were women, and the proportion of those reporting at least a high school diploma significantly increased over time (from 9.9% in 2003 to 17% in 2016). About 45% of those interviewed were married, and the proportion who had never married or who were separated/divorced increased by about 1 percentage point (from 8.2 to 9.2%). The prevalence of chronic limitations also increased (from 36 to 54%), as did the percentage of those reporting adequate economic household resources (from 60 to 66%). The proportion of individuals having at least one child living close to them remained stable over the 13 years considered (at 58%) while the figure for respondents having (all of the) child(ren) living at a distance increased (from 24 to 27%).

Finally, while the number of respondents living in medium-large areas slightly increased over time, those living in small localities decreased from 33% in 2003 to 30% in 2016.

### 4.2. Trends in the Use of Social, Health and Private Services among Older Individuals in Italy

Table 2 reports AMEs and robust standard errors from logistic regression models of the use of social, health and private LTC services among Italians aged 75 or over in 2003 and 2016.

With regard to the main independent variables (i.e., municipality size and region), older people living in medium and small municipalities were about 2 percentage points more likely to have used public social home care services in the last year compared to those living in metropolitan areas. At the same time, they were significantly more likely to have used health services, especially in areas with fewer than 10,000 inhabitants (+4 percentage points; *p* < 0.05). However, these results were only significant in 2016; in 2003, no significant differences by local authority size were found. Finally, in 2003, estimates show that individuals aged 75 and over living in small areas were about 2 percentage points less likely to have used private care in the last 12 months, compared to those living in metropolitan areas; this difference had lost its significance in 2016.

Compared to older people living in the South of Italy, those living in the Northwest were more likely to have used social services in 2003 (AMEs = 0.03; *p* < 0.01). At the same time, those living both in the Northwest and in the Northeast, as well as those living in the Centre of Italy were significantly more likely to have used health services in 2003. Finally, results show that older people living in the Centre were also about 2 percentage points more likely to have used private home care services in 2003.

Regarding socio-economic and demographic characteristics of the respondents, those with higher educational attainments were less likely to have used health services in 2016 (AMEs = −0.05; *p* < 0.01) but more likely to have used private home care services in 2003 (AMEs = 0.04; *p* < 0.01). Findings also show that widows or widowers were about 2 (in 2003) to 6 (in 2016) percentage points more likely to have used private home care services in the last year, compared to those living in a couple.

As expected, older people with chronic conditions were more likely to have used public social and health home care services or private home care services. However, this probability slightly decreased in the 13 years considered (−1/2 percentage points). Finally, compared to the childless, those who had a child, especially living close to the respondent, had a significantly lower probability of having used private home care services in the previous year of about 8 percentage points.

## 5. Discussion

According to our analysis, those living in small towns had a higher probability of using social and health home care, compared to those living in metropolitan areas (+2% and +4%, for social and health services, respectively, in 2016). This is the first time that an Italian study has investigated these particular territorial differences. Interestingly, these differences contrast with the current territorial distribution of public resources since the Great Recession, which result in small municipalities having less to spend than metropolitan areas on social home care., This raises questions on territorial inequality in resource distribution beyond the well-known North–South divide. Taking into account the observed effects of the pandemic period, such shortcomings in the LTC system could become very dangerous in the near future.

Our results show how the percentage of Italian people aged 75 and over who accessed home health care services increased from 7.7% in 2003 to 9.7% in 2016, whereas public home social care service use slightly decreased from around 3% in 2003 to 2% in 2016, confirming that, during the Great Recession, regions and municipalities restricted the budgets allocated to home care services specifically to those most in need (i.e., those with severe health problems), who were often the oldest [18,35]. Trends are comparable to the administrative data described in the background section, but higher proportions were found since people aged 75 and over were considered (rather than 65 years and over in administrative data). At the same time, older people employing private care workers increased from 5.8% in 2003 to 7.9% in 2016. Older people with children living in close proximity were less likely to hire private care (−8% compared to childless people), as previously found in literature [8,19,23,46].

People living in the Northern and Central regions of the country had a higher probability of accessing public home health care services compared to those living in Southern areas (+3% in Central and Northeast areas, +4% in Northwest areas in 2016): an expected result, considering the higher budget of northern Regions allocated to social and health services [18,21,33,35]. This result may pose questions on the equity of the current Italian LTC system, which leaves the frailest people living in the poorest regions with even fewer public resources for LTC.

Focusing on municipality size, around one-third of respondents in the sample were living in municipalities with fewer than 10 thousand inhabitants. Those living in small towns had a higher probability of using social and health home care, compared to those living in metropolitan areas (+2% and +4% for social and health services, respectively, in 2016). This result is in contrast with previous literature that shows how small municipalities spent less in social home care by comparison with metropolitan areas, after the Great Recession [21]. It is noteworthy that there were no significant differences in 2003, i.e., pre-Recession. A possible explanation is that access to municipal and regional (NHS) services for those living in small/medium towns is more easily compared to those living in bigger cities: are small municipalities more efficient in organising home care services? Additionally, it is possible that there is a rural–urban gradient in attitudes towards institutionalization [25,26]; perhaps older people living in small communities are less inclined to move to an institution and prefer to stay at home, closer to their social network [26; 30]. In this context, it is indeed interesting to underline two results from the model estimations: first, differences in employing private carers disappear in 2016 (the probability was significantly lower in 2003 for those living in small municipalities); second, proximity of children significantly lowers the probability of employing private carers (−8% if the closest child lives less than 1 km away). A greater attachment to the home town and proximity to the social and family network may encourage dependent older people living in small municipalities to stay at home and use municipal/regional home care services jointly with some additional private home care, rather than move to a nursing home. Finally, rural cultural norms can play a role in such decisions [47]. A combination of these factors could lead those living in small/medium towns to avoid moving to an institution, compared to those living in big cities, and consequentially, explaining a higher propensity to use public home care.

In both instances (better efficiency of small towns or greater resistance of rural older people to moving to a nursing home), new and greater investments in home care services are urgently needed at the national level, but such investments are even more crucial in southern areas and small towns, that suffer lower budget resources and (in the case of smaller towns) present higher propensity to use home care services. This is a lesson for the post-COVID-19 pandemic period: a new design and greater investments in the national public home care service system are undelayable, in particular to rebalance Italian geographic disparities (cf. [27,31]).

Conversely, lower home care service use in metropolitan areas may suggest a need for reviewing their structure, to increase accessibility and proximity to older people, disseminating information on the available services and strengthening their integration and coordination with other sources of support (cf. [48]).

In both the cases, two concrete policy changes are urgent:(1)The fragmentation of the LTC system needs to be harmonized and possibly integrated in order to overcome territorial inequalities in resource allocations.(2)Home health care, Home social care and Institutionalization should have integrated access points in order to offer older people and their relatives a real possibility of choosing the services that best fit their needs.

Several limitations affect this study. Firstly, as in all the Italian population surveys, older people living in nursing homes are not included in the sample design, limiting our knowledge of preferences among older people dealing with severe disabilities. Secondly, given the cross-sectional nature of the surveys it is impossible to study causal relations between variables, but merely to examine statistically significant associations. Last, but not least, municipality size is not an ideal indicator for clustering municipalities for the services they provide. Different municipalities may provide services for different reasons independent of population size.

Nevertheless, the perfect comparability of the two surveys, jointly with the powerful sample size, and the use of municipality size as a proxy for local administration of care services, are strong advantages of this study.

## 6. Conclusions

This study focused on home care services as alternatives to institutionalization, the weaknesses of which were dramatically exposed during the COVID-19 pandemic. Although this study considers the period 2003–2016, the current Italian public care system has not witnessed any substantial change since that time, as confirmed by administrative data and literature cited in the background section. Therefore, the results and conclusions may be considered still relevant. Nevertheless, exploring the impact of COVID-19 on the LTC system will be absolutely crucial once new data become available. Consolidating home services proved to be a winning strategy during the pandemic, since it may have prevented many more casualties of older people resident in nursing homes. The results confirm the inadequacy of the Italian public home care system but, more than this, unveil interesting differences among municipalities by size and geographic location. Whereas northern areas and bigger municipalities can invest more resources in home care services due to their more substantial budgets, older people living in small municipalities seem more likely to use public home care services than their peers living in metropolitan areas. Better efficiency of small municipalities or stronger resistance of rural older people to moving to nursing homes are potential explanations, but both call for a substantial rebalance of home care investments among regions and municipalities. Moreover, deep disparities in economic resources among Italian Regions call for a more homogenous direction of the system at national level, increasing the integration between health and social home care service, at the national, regional and municipal level.

## Figures and Tables

**Table 1 ijerph-19-12796-t001:** Descriptive statistics.

Independent Variables	2003	2016
	%	
	N = 4014	N = 3314
Territorial sizes	2003	2016
Metropolitan area	26.15	27.15
>10,000 inh.	40.62	43.03
<10,000 inh.	33.23	29.82
Gender		
Women	62.62	60.27
Educational attainment		
Middle-High	9.94	17.06
Marital status		
Married	43.19	47.16
Never married/Separated/Divorced	8.21	9.23
Widow	48.60	43.60
Macro-region		
Northeast	25.48	28.24
Northwest	21.46	19.82
Centre	21.49	21.18
South	21.59	20.59
Islands	9.98	10.16
Chronic diseases	36.03	54.56
Adequate economic resources	60.65	66.11
Children		
No children	16.06	14.02
Having a child living > 1 Km	24.97	27.85
Having a child living < 1 Km	58.97	58.12
Dependent Variables	%	
Public Home Social Care	2.93	2.14
Public Home Health Care	7.67	9.73
Private care	5.78	7.87

Note: Weighted data; Source: own elaborations on ISTAT Data Sample. Families, Social Subjects and life cycle (FSS), ISTAT, 2003 and 2016. The significance level for all analyses was set at *p* < 0.05.

**Table 2 ijerph-19-12796-t002:** Average marginal effects and robust standard errors from logistic regression models of social, health and private care receipt.

	Social Care-Logit		Health Care-Logit		Private Care-Logit	
VARIABLES	2003	2016	2003	2016	2003	2016
	AMEs (S.E)					
>10,000 inh. (ref. Metropolitan area)	0.01	0.02 ***	0.00	0.03 **	−0.01	0.02
	(0.01)	(0.01)	(0.01)	(0.01)	(0.01)	(0.01)
<10,000 inh. (ref. Metropolitan area)	0.01 *	0.02 ***	0.00	0.04 **	−0.02 **	−0.01
	(0.01)	(0.01)	(0.01)	(0.01)	(0.01)	(0.01)
Women	−0.00	−0.00	0.00	−0.03 **	0.01	−0.01
	(0.01)	(0.01)	(0.01)	(0.01)	(0.01)	(0.01)
Middle-High educational attainment	−0.03 *	−0.01	0.01	−0.05 ***	0.04 ***	0.01
	(0.02)	(0.01)	(0.01)	(0.02)	(0.01)	(0.01)
Never married/Separated/Divorced (ref. Married)	0.02	−0.00	0.02	−0.01	0.00	0.01
	(0.01)	(0.01)	(0.02)	(0.02)	(0.01)	(0.01)
Widow (ref. Married)	−0.00	0.01	−0.00	0.02	0.02 **	0.06 ***
	(0.01)	(0.01)	(0.01)	(0.01)	(0.01)	(0.01)
Northeast (ref. South)	−0.01	0.00	0.04 ***	0.03 *	0.01	0.01
	(0.01)	(0.01)	(0.01)	(0.02)	(0.01)	(0.01)
Northwest (ref. South)	−0.00	0.03 ***	0.03 ***	0.04 ***	0.00	0.00
	(0.01)	(0.01)	(0.01)	(0.02)	(0.01)	(0.01)
Centre (ref. South)	−0.01	−0.01	0.05 ***	0.03 **	0.02 **	0.00
	(0.01)	(0.01)	(0.01)	(0.02)	(0.01)	(0.01)
Islands (ref. South)	0.02 *	0.02	0.01	0.00	0.02 *	0.03 *
	(0.01)	(0.01)	(0.01)	(0.02)	(0.01)	(0.02)
Chronic diseases	0.04 ***	0.02 ***	0.11 ***	0.10 ***	0.11 ***	0.08 ***
	(0.01)	(0.01)	(0.01)	(0.01)	(0.01)	(0.01)
Adequate economic resources	−0.00	−0.01	−0.01	0.00	−0.00	−0.01
	(0.01)	(0.01)	(0.01)	(0.01)	(0.01)	(0.01)
Having a child living > 1 Km	−0.00	−0.02 *	−0.02	−0.02	−0.01	−0.05 **
	(0.01)	(0.01)	(0.01)	(0.02)	(0.01)	(0.02)
Having a child living < 1 Km	−0.02	−0.01	−0.00	−0.01	−0.02 *	−0.08 ***
	(0.01)	(0.01)	(0.01)	(0.02)	(0.01)	(0.02)
Observations	4014	2955	4014	2953	4014	2982
Standard errors in parentheses, *** *p* < 0.01, ** *p* < 0.05, * *p* < 0.1						

Source: own elaborations on ISTAT Data Sample. Families, Social Subjects and life cycle (FSS), ISTAT, 2003 and 2016.

## Data Availability

Third Party Data. Data were obtained from Italian Statistical Institute (ISTAT) and are available from the authors with the permission of ISTAT.

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
