# Peer review of "Trends in the Use of Home LTC Services in Large, Medium and Small Municipalities in Italy: Lessons for the Post-COVID-19 Reappraisal"

_ijerph, 2022, doi:10.3390/ijerph191912796_

Round 1

Reviewer 1 Report

Dear authors: There needs to be another review of the language. Some of the sentences are too long and there are misspelled words. Plagiarism, you have used another persons statistics, this is a sort of plagiarism, you are simply interpreting the results, so i recommend for you to add another level of methodology. Have a focus group, and invite some of the seniors to give you input concerning your research questions and purpose of study. You give no real examples of recommendations, you should give the health system some specific reasons of what they need to do based on your research. 

Reviewer 2 Report

Congratulations on the choice of topic and scientific quality of the article.

Reviewer 3 Report

Abstract - do not use abbreviations in abstract 

Introductions - quite wide explained topic, but I didn't got impression what is new in this manuscript and what is scientific contribution of your work.

Discussion -start with your study aim and highlite your main finding.  
